# PSEN2 Mutations May Mimic Frontotemporal Dementia: Two New Case Reports and a Review

**DOI:** 10.3390/biomedicines12081881

**Published:** 2024-08-17

**Authors:** Anxo Manuel Minguillón Pereiro, Beatriz Quintáns Castro, Alberto Ouro Villasante, José Manuel Aldrey Vázquez, Julia Cortés Hernández, Marta Aramburu-Núñez, Manuel Arias Gómez, Isabel Jiménez Martín, Tomás Sobrino, Juan Manuel Pías-Peleteiro

**Affiliations:** 1Servicio de Neurología, Hospital Clínico Universitario Santiago de Compostela, Travesía de Choupana, 15706 Santiago de Compostela, Spainjuan.manuel.pias.peleteiro@sergas.es (J.M.P.-P.); 2Fundación Pública Galega de Medicina Xenómica, Hospital Clínico Universitario Santiago de Compostela, Rúa da Choupana, 15706 Santiago de Compostela, Spain; 3Centro de Investigación Biomédica en Red de Enfermedades Raras (CIBERER-U711), 15706 Santiago de Compostela, Spain; 4NeuroAging Laboratory (NEURAL) Group, Clinical Neurosciences Research Laboratory (LINC), Health Research Institute of Santiago de Compostela (IDIS), Hospital Clínico Universitario, 15706 Santiago de Compostela, Spain; alberto.ouro.villasante@sergas.es (A.O.V.);; 5Centro de Investigación Biomédica en Red en Enfermedades Neurodegenerativas (CIBERNED), Instituto de Salud Carlos III, 28029 Madrid, Spain; 6Servicio de Medicina Nuclear, Sección de Sistema Nervioso Central, Hospital Clínico Universitario Santiago de Compostela, Travesía de Choupana, 15706 Santiago de Compostela, Spain; 7Unidad de Neuropsicologia Clínica, Hospital Clínico Universitario Santiago de Compostela, Travesía de Choupana, 15706 Santiago de Compostela, Spain

**Keywords:** Alzheimer’s disease, frontotemporal dementia, presenilin 2, phenocopies, white matter hyperintensities

## Abstract

Background: Monogenic Alzheimer’s disease (AD) has severe health and socioeconomic repercussions. Its rarest cause is presenilin 2 (*PSEN2*) gene mutations. We present two new cases with presumed PSEN2-AD with unusual clinical and neuroimaging findings in order to provide more information on the pathophysiology and semiology of these patients. Methods: Women aged 69 and 62 years at clinical onset, marked by prominent behavioral and language dysfunction, progressing to severe dementia within three years were included. The complete study is depicted. In addition, a systematic review of the PSEN2-AD was performed. Results: Neuroimaging revealed pronounced frontal white matter hyperintensities (WMH) and frontotemporal atrophy/hypometabolism. The genetic study unveiled *PSEN2* variants: c.772G>A (p.Ala258Thr) and c.1073-2_1073-1del. Both cerebrospinal fluid (CSF) and experimental blood biomarkers shouldered AD etiology. Conclusions: Prominent behavioral and language dysfunction suggesting frontotemporal dementia (FTD) may be underestimated in the literature as a clinical picture in *PSEN2* mutations. Thus, it may be reasonable to include *PSEN2* in genetic panels when suspecting FTDL. *PSEN2* mutations may cause striking WMH, arguably related to myelin disruption induced by amyloid accumulation.

## 1. Introduction

Alzheimer’s disease (AD) due to mutations in a single gene is transmitted to descendants with an autosomal dominant inheritance (ORPHA: 1020) and has a penetrance close to 100%. ORPHANET estimates its prevalence at around 1–9/100,000 cases [1].

Considering the age of symptomatic onset as the classifier parameter, AD can be divided into early-onset type (EOAD) and late-onset type (LOAD), establishing the borderline at 65 years of age. Approximately 2–10% of the total number of patients belong to the first group. Nevertheless, autosomal dominant forms, with an estimated proportion of less than 1% overall, are calculated to cause only 5–10% of EOAD [2].

These monogenic mutations have been described in three genes: amyloid beta precursor protein (*APP*) [3], presenilin-1 (*PSEN1*) [4], and presenilin-2 (*PSEN2*) [5], all of them with either direct or indirect implications in the so-called amyloid cascade.

Autosomal dominantly inherited AD caused by mutations in the *PSEN2* gene is the least frequent of all monogenic AD. The symptomatic phase of AD-PSEN2 can occur at any age, but its penetrance is not complete, standing at around 95%. Pathogenic mutations in these three genes produce increased levels of Aβ40 and Aβ42 peptides. Aβ42 is also the most common peptide in the typical AD extracellular plaques, mainly due to its high aggregation tendency. The *APP* gene encodes a transmembrane protein of the same name, which causes the substratum to increase [6]. In *PSEN1* and 2 mutations, pathogenicity arises through alterations in the complex that cleaves the APP protein, specifically in the catalytic site of the γ-secretase. These changes will lead to abnormal proteolysis of APP, causing an accumulation of Aβ peptides, with Aβ42 being the most abundant. In this regard, Aβ42 induces multiple toxic effects and stimulates neuroinflammation either directly or by initiating the neurodegenerative pathway [7].

From a clinical perspective, previous descriptions highlight the presence of classical AD features such as progressive memory impairment and disorientation [8,9]. Congruently, neuroimaging outlines emphasize that brain regions most commonly affected in *PSEN2* patients hold no major differences, with parietal and temporal lobes being the main targets of neurodegeneration, hypoperfusion, and therefore hypometabolism. However, an irregular involvement of the frontal lobes has been noted in some cases [8]. Overall, the molecular, clinical, and neuroimaging characteristics of monogenic AD patients are still largely unknown. Here, we present two new cases of presumed PSEN2-AD with unusual clinical and neuroimaging findings in order to provide more information on the pathophysiology and semiology of these patients [10].

## 2. Detailed Case Description

### 2.1. Materials and Methods

#### 2.1.1. Clinical and Neuropsychological Workup

The patients were evaluated at the Clinical University Hospital of Santiago de Compostela (Galicia, Spain).

Clinical data related to initial symptoms, disease progression, family history, and other medical history were collected. A thorough neurological examination was conducted by a professional neurologist. A longitudinal follow-up was conducted yearly to observe disease progression.

A standardized battery of tests was administered by a neuropsychologist every two years, measuring cognitive function in the domains of memory, executive function, language, and behavior.

Global cognitive screening measures included the Mini-Mental State Examination (MMSE) [11], the Montreal Cognitive Assessment (MoCA), and the Frontotemporal Lobar Degeneration (FTLD)—Clinical Dementia Rating Scale [12]. Visuo-spatial function was evaluated using the Rey–Osterrieth complex figure test and simple figures (pentagons, etc.). Language was measured using the Boston Aphasia Severity Scale [13] and also with other specific tests [14]. Anterograde episodic memory was measured using the Memory Impairment Screen [15]. Behavioral dysfunction was evaluated using the Frontal Behavior Inventory and the Neuropsychiatric Inventory (NPI) [16]. The ability to perform daily living activities was assessed using the Activities of Daily Living Scale (ADL) [17]. The test results for which an evolution over the years is available are summarized in Table 1.

#### 2.1.2. Structural and Functional Neuroimaging Tests

Structural and functional neuroimaging were performed in both patients at the Clinical University Hospital of Santiago de Compostela (Galicia, Spain).

FDG-PET was conducted using a dose of 189 MBq. A brain metabolism study was performed with 3D mode acquisition and attenuation correction using low-dose brain CT. Magnetic resonance imaging (MRI) was obtained with a Philips Ingenia Ambition 1.5T MRI equipment; protocol: 3D T1, 3D T2, T2-FLAIR, susceptibility sequences, and diffusion sequences.

#### 2.1.3. Laboratory Study (Including AD Biomarkers)

Fasting blood samples were obtained in the morning (<10 a.m.). Briefly, blood samples were collected in chemistry test tubes for obtaining plasma. Both CSF and blood samples were centrifugated at 1700× *g* for 15 min at 4 °C. Then, plasma was collected and centrifugated at 2000× *g* for 5 min at 4 °C. Each aliquot of 0.5 mL was stored at −80 °C until required for analysis.

Standard biochemistry, hematology, and coagulation tests were assessed in the central laboratory of the Clinical University Hospital of Santiago de Compostela. CSF and plasma Aβ40, Aβ42, and p-tau181 as well as CSF total tau protein (t-tau) were quantified with the Lumipulse automated platform (Fujirebio, Europe, sourced from PA (USA)) following the provider’s instructions, in line with Global Biomarker Standardization [18,19] in the Clinical Neurosciences Research Laboratory of Health Research Institute of Santiago de Compostela (IDIS). For each sample, Aβ1–42, Aβ1–40, t-tau, and p-tau181 were quantified simultaneously in the same run immediately after the first freeze–thaw cycle of each sample using pristine aliquots containing 500 μL of CSF/plasma [20].

#### 2.1.4. Genetic Tests

As for genetic testing, in case A, SureSelect (Agilent) probe hybridization and NGS (Ion Proton, sourced from MA (USA)) sequencing of coding exons and flanking regions up to 10 base pairs of a panel of 18 dementia-related genes were carried out. Bioinformatic analysis used the tools TMAP4.4-11, TVC5.0-7, GATK 3.4-46, ANNOVAR 2015Jun17, and IGV 2.3. All genes had 100% coverage in the analysis (calculation on the target region, with a depth of at least 30×). The mean reading depth for this sample was 241×.

For case B, NGS sequencing (Illumina NextSeq 500) was carried out for the entire coding region and the surrounding intricate regions of the *APOE*, *APP*, *C9orf72*, *CHD10*, *CHMP2B*, *CSF1R*, *FUS*, *GRN*, *HNRNPA1*, *HNRNPA2B1*, *HTRA1*, *ITM2B*, *MAPT*, *NOTCH3*, *NPC1*, *NPC2*, *PLA2G6 PRKAR1B*, *PRNP*, *PSEN1*, *PSEN2*, *SNCA*, *SNCB*, *SQSTM1*, *TARDBP*, *TBK1*, *TREM2*, *TUBA4A*, *UBQLN2*, and *VCP*. Data analysis used the following computer tools: fastp v0.19.7, bwa 0.7.17-r1188, GATK v3.8-0, GATK v4.1.0.0, Picard 2.18.27, BEDTools v2.27.1, SAMtools 1.9, and ANNOVAR 2018Apr16. All of them using the latest software available. The mean reading depth of the test for this sample was 565×.

Visualization was performed using IGV 2.3 Human genome version hg19. Interpretation and classification of variants was performed according to recommendations of the American College of Medical Genetics and Genomics (PMID: 25741868) and ACGS Best Practice Guidelines for Variant Classification in Rare Disease 2020. The description of the genetic variants reported follows the recommendations of the HGVS (Human Genome Variation Society).

### 2.2. Case A

#### 2.2.1. Clinical Course

Patient A was a right-handed, 62-year-old Caucasian woman at the time she was admitted to our hospital. She was born and raised in Galicia (northwestern Spain). She had studied for ten years and later worked as a housewife and in a factory of canned food. She was married and had two healthy sons. She had no personal history of vascular risk factors or other relevant medical history. Her father had a non-specified cognitive decline and eventually severe dementia at the age of 85. Her mother had died at age 35 from a bacterial infection, and other information regarding her mother’s family was lacking. She had two older siblings: a healthy brother aged 72 and a sister aged 70 diagnosed with Parkinson’s disease without associated cognitive impairment.

Two years before, at the age of 60, her family observed initially subtle behavioral changes inconsistent with her previous personality, such as self-neglect and lack of care about personal hygiene. When confronted with this particular topic, she had disproportionate anger bursts. She gradually developed hyperphagia, with a special preference towards sweets and French fries, and experienced a congruent weight gain. Hyperorality also developed in the form of a peculiar stereotypical movement of continuously touching her teeth with her fingers.

At the same time, she also developed a language impairment that started as difficulty in word finding and rapidly evolved towards a reduced fluency, a “telegraphic style”, and an overuse of short answers and set phrases.

Approximately two years after clinical onset, she was admitted to our hospital. A neuropsychological examination revealed an abnormal cognitive status with an executive, language (including comprehension), and memory dysfunction, further developed in Table 1, which also describes quantitative evolution over time. This neuropsychological assessment was significantly limited by the severe linguistic impairment.

Neurological examination revealed noted regressive reflexes (bilateral grasp, right palmomental, rooting, and glabellar reflexes) but was otherwise unremarkable. She had normal cranial signs and normal limb motor, reflex, sensory, and cerebellar function. There were no signs of meningeal irritation or involuntary movements.

Three years after clinical onset, she developed inappropriate laughing, shouting, and a marked disinhibition, shown, for example, in walking naked through her house before taking a shower. A gradual decline in instrumental activities of daily living was also noted, with progressive abandonment of house chores.

Four years after clinical onset, she became incontinent of urine and feces and completely dependent for basic activities of daily living except for eating, which she achieved with a spoon or more often with her bare hands.

Significantly, she became very affectionate, often kissing familiar people (including the doctor) of warmly holding other people’s hands against her chest. Language became limited to single words, although, paradoxically, she occasionally sang short, repeated songs. Hyperphagia persisted.

Within five years, she lost the ability to walk unaided. Neurological examination revealed an asymmetric parkinsonism, with cogwheel rigidity in her right upper extremity and global bradykinesia.

Six years after clinical onset, she became bed-ridden and mute but retained a social smile and sometimes laughed in response to repeated stimulus.

The patient died at age 68, approximately 8 years after clinical onset.

#### 2.2.2. Neuroimaging Tests

When performed, PET-FDG neuroimaging showed a predominant frontal lobe impairment, together with metabolic preservation of other regions classically altered in AD. In the MRI study, moderate atrophy with frontal predominance was observed as well as WMH in the same regions (Figure 1).

#### 2.2.3. Genetic Testing

The genetic study did not find any DFT-related mutations, including *APP*, *CHMP2B*, *FUS*, *GRN*, *HNRNPA1*, *HNRNPA2B1*, *ITM2B*, *MAPT*, *PRNP*, *PSEN1*, *SNCA*, *SNCB*, *TARDBP*, *TIMM8A*, *TREM2*, *UBQLN2*, and *VCP* [21]. The result with respect to *C9orf72* in this patient was negative regarding RP-PCR expansion. Noticeably, the genetic study revealed a rare novel heterozygous point mutation at exon 8 (NM_000447.3:c.772G>A) of the PSEN2 gene, which causes a change from alanine to threonine at codon 258 (p.Ala258Thr) (Figure 2 and Figure 3).

This has not been described in patients, so it has no associated phenotypic data. In the general population, it is found in low frequency (in Europeans 0.02%; rs148238688), and the in silico algorithms indicate its possible pathogenicity (PHRED-scaled CADD score 26.8; REVEL 0.84). This variation was previously considered heterogeneously, for example, as benign on the Alzforum website (last updated: 22 February 2022; https://www.alzforum.org/mutations/psen2-a258t, access date: 10 February 2024), mainly due to its relatively high frequency in controls; on the other hand, ClinVar contains an entry for this variant (Variation ID: 652702) as classified of uncertain significance. In the same codon, the variant c.773C>T (p.Ala258Val) was found in a Japanese individual diagnosed with probable AD and a family history of dementia [22]. With the data currently available, c.772G>A should be considered a variant of unknown significance, potentially pathogenic.

### 2.3. Case B

#### 2.3.1. Clinical Course

Patient B was a right-handed, 69-year-old Caucasian woman, also born in Spain. She had studied until the age of 14. She later worked as a waitress in hotels in Switzerland, Austria, and eventually back in Spain. She was married for a short period of time and divorced at age 40, leaving no descendants. Her medical history was unremarkable.

Both parents had died, and she had no siblings. Both her mother and her maternal grandmother had developed cognitive impairment starting before the age of 70. Information regarding his father was limited, as she was the daughter of a single mother. There was otherwise no known family history of parkinsonism, motor dysfunction, or any other neurodegenerative diseases. The caregivers were indeed not relatives but instead a couple of close friends.

At the age of 68 years, the patient’s friends observed prominent personality changes. She became introverted and lacked initiative. She often reported she was “not feeling well”, even to unfamiliar persons (e.g., in a grocery store). Notably, in one occasion, with justification, she accused some close friends of “stealing her things”. There was no evidence self-neglect; on the contrary, she developed an obsession with personal hygiene. Her friends also observed diminished retention of new information and constant withdrawal from conversations, plainly answering, “I don’t know”. A prodromal weight loss of around 10 kg was also noted.

Approximately one year after clinical onset, she was admitted to our hospital. A neuropsychological examination revealed an abnormal cognitive status with executive, language, and memory dysfunction, which is further described in Table 1. Neurological examination was otherwise unremarkable. Only a mild improvement in her lack of initiative was observed after treating with an anti-depressive drug (escitalopram).

Two years after clinical onset, her condition worsened. Caregivers reported impulsive behavior and irascibility. She spoke even less, using only short sentences. Worsening was also noticed in the functional domain, with a sharp decline in instrumental activities.

Within three years, the patient experienced a further functional decline, being partially dependent for basic daily living activities and intermittently incontinent of urine. Significantly, she developed pathological laughing, which was triggered by minor stimulus. On the other hand, she easily lost her temper with no apparent reason.

Compared with the first follow-up, regressive reflexes were observed (bilateral grasp and palmomental, with a right predominance). The neurological examination remained otherwise unremarkable.

Four years after clinical onset, there was a further decline regarding basic activities of daily living. She became incontinent of urine and feces. Pathological laughing persisted, accompanied by an inappropriate affectivity (e.g., attempting to kiss the doctor). The neurological examination revealed a subtle right-limb hypertonia, not associated with tremor, in addition to previous findings.

Within approximately five years, she became bed-ridden. She also presented with focal seizures, which were effectively treated with levetiracetam. Language dysfunction evolved towards mutism, interrupted only by loud, persevering moans and alliterations (“lalalala”). At the time of writing, the patient is still alive.

#### 2.3.2. Neuroimaging Tests

Noticeably, the MRI study once again showed confluent hyperintense lesions in the white matter of both frontal lobes. In FDG-PET and FDG-PET with the SPM technique images, frontal cortical hypometabolism was observed with left-side-predominant involvement; no metabolic abnormalities were seen in the occipital, parietal, or temporal lobes (Figure 4). SPM (statistical parametrical mapping) performs an analysis by comparing, one-by-one, the different voxels into which the patient’s brain is previously divided with the voxels of an unaffected control brain. The processing of these data allows us to obtain the mapping of the most altered regions coinciding with the data provided up to now.

#### 2.3.3. Genetic Testing

Once more, although the clinical diagnosis suggested FTD, i.e., in this case, a behavioral variant, the genetic study intriguingly revealed a heterozygous deletion of two nucleotides at the acceptor splice site of the intron 11 of the PSEN2 gene NM_000447.3:c.1073-2_1073-1del, presumably resulting in exon 12 skipping during gene processing (Figure 2). This mutation was formerly reported in a patient with an ALS diagnosis [23]. Another frameshift variant in this position (c.1073-2delA; p.G359Lfs*74) was also reported in two patients with mild cognitive impairment and Alzheimer’s disease, in which expression studies showed reduced PSEN2 [23]. In summary, with the available pieces of evidence, the c.1073-2_1073-1del variant is considered likely pathogenic. Integrative Genomic Viewer (IGV) representations of both mutations found are shown in Figure 3.

In addition, case B was found heterozygous for rs429358. This result corresponds to the APOE E3/E4 genotype. Case B is compound heterozygous for normal alleles in C9orf72, so she has no carried pathological expansion. The APOE E4 allele is associated with an increased risk of developing Alzheimer’s disease (OR = 2–3 in heterozygosis; OR = 14.9 in homozygosis [24]), although the presence of the APOE E4 allele is not necessary or sufficient to develop Alzheimer’s disease. Likewise, the presence or absence of the APOE E4 allele does not exclude that the individual carrying it could have another genetic alteration causing dementia.

### 2.4. AD Blood and CSF Biomarkers

In order to support a diagnosis of Alzheimer’s disease, an attempt was made to analyze AD CSF and also serum biomarkers. In case A, after signing the informed consent, lumbar puncture was performed following the required aseptic procedures and with an absence of complications, with the removal of four sample tubes. In case B, the patient’s legal guardian did not consent to the lumbar puncture; however, it was possible to collect a blood sample for the determination of AD biomarkers in plasma, which were also determined for case A.

So far, no correlation has been determined between the biomarkers analyzed in CSF and plasma. However, different studies have determined higher levels of these biomarkers in AD patients compared to healthy people [25,26,27]. In addition, recent work has determined that p-tau levels greater than 2 pg/mL and an Aβ 42:40 ratio less than 0.083 could be considered AD biomarkers for more exhaustive tests (Preprint: https://www.medrxiv.org/content/10.1101/2023.04.20.23288852v1, access date 9 September 2023—doi included in link).

Overall, both serum and CSF biomarkers support, in both cases, an AD etiology. Regarding the non-decreased amyloid 1–42 values in case A, we hypothesize that this is a pathological amyloid-hyperproducing patient [28,29]. The Aβ 42:40 ratio in CSF is close to the pathological threshold, and we believe that this is due to the massive quantity of amyloid 1–40 (10,693), and equally so in the blood sample (318.71), where a clearly pathological value was obtained in the Aβ 42:40 ratio.

Biomarker results in CSF and serum are presented in Table 2 and Table 3.

## 3. Discussion

Given the remarkable results of the genetic testing in two patients with clinical and neuroimaging studies compatible with FTD variants (non-fluent primary progressive aphasia being the most probable clinical diagnosis in case A and behavioral frontotemporal dementia in case B), a systematic review of AD and *PSEN2* was carried out, prioritizing the clinical perspective (Figure 4).

The search was performed in the MedLine and PubMed servers and databases. It was carried out using several simple but focused search terms to gather as much information as possible about the search terms but focused on gathering the maximum information about familial Alzheimer’s disease caused by or related to the gene under study. In addition, some Boolean operators such as “AND” and/or “OR” were also applied to manage the search: [(“Alzheimer´s disease” OR “AD”) AND (“Presenilin 2” OR “*PSEN 2*”)].

In order to perform an accurate systematic review on the relationship between Alzheimer’s disease and presenilin 2, we selected studies with high statistical value, such as meta-analyses, multicenter studies with large samples, systematic reviews, and clinical trials. Although they were large studies, in some of them, the data extracted focused only on clinical data from specific cases, especially in an attempt to compose a phenotype common to *PSEN2* mutations.

Nowadays, it is an established idea that mutations in presenilins associated with familial AD are directly or indirectly involved in the increase in the APP-to-Aβ42 pathway as well as in the increase in Aβ42 concentration itself. What is still not completely clear is whether this is due to, on the one hand, an increase in the total activity of γ-secretase (gain), which seems to have received greater support from the current literature so far [30], or, on the other hand, to a decrease in its function (loss), with some reports also subscribing this hypothesis [31].

Concerning the development and clinical debut itself, no consensus has been reached on what would be a paradigmatic picture in the case of *PSEN2*. However, there are some common features: The main symptom of the disease debut is progressive and insidious memory impairment (reaching 88% in some series) and in the form of disorientation (more than 36% in the same study) [8,9]. In the German Volga cohort (N141I), it was described that the development of the disease usually ended in mutism, rigidity, and prostration [32]. In addition, multiple sources allude to the considerable presence of symptoms of the psychiatric sphere “BPSD” (Behavioral and Psychological Symptoms of Dementia), with depression being the most frequent (more than 27% in Shea et al.) as well as apathy (21% in the same study), delusions (more than 12%), and hallucinations (9%), all of the visual subtype [9]. Another characteristic clinical feature is the relatively high frequency of seizures in these patients, which are overall associated with monogenic AD (reaching more than 47%) [33]. In the case of *PSEN2*, the percentage ranges from 15.2% [9] to almost 43% [33]. In fact, one-third of these subjects suffered at least one episode of seizures during the evolution of their disease [32,34].

The presence in PSEN2-AD patients of epileptic seizures during the evolution of the disease is not only a descriptive record of this cohort of patients, but there have been several attempts to determine at the biological level the mechanism for the malfunction of this protein to give rise to the susceptibility of epileptogenic circuits [35] (playing an important role in Ca^2+^-mediated regulation and signaling).

When in such a situation that the disease also leads to cerebral amyloid angiopathy (CAA), the risk of seizures is increased by the acute or peri-acute phase of intracranial hemorrhages and, in the long term, by the irritability of the superficial siderosis and the presence of cortical amyloid itself. In a recent study published in *Epilepsia*, the presence of focal or disseminated cortical superficial siderosis was associated with an increased risk of epilepsy in all patients with CAA, and the association with inflammatory CAA and seizures was especially significant in the whole sample. In these patients, the most frequent seizure type was focal (81.3%), with a non-negligible risk of status epilepticus. In our cases, case A presented focal epileptic seizures without impaired awareness, with some of them progressing into bilateral tonic-clonic seizures.

Regarding the role of mutations that give rise to familial Alzheimer’s disease and, more particularly, those of *PSEN2* in the development of cerebral amyloid angiopathy (CAA), there are several reports of patients in whom intracranial hemorrhages were described, mostly in the cohort with the p.Asn141Ile mutation [36]. In none of the cases we present, which had susceptibility-weighted imaging (SWI) sequences in MRI, were data of superficial siderosis or macrohemorrhages observed. In case A, the presence of microhemorrhages was detected in both hemispheres.

We want to again highlight the fact that, in both cases, the early age of onset, the clinical features (showing a strong behavioral component, involvement of executive areas, decision making, and social adequacy as well as language impairment) and also neuroimaging—where the regions with the greatest metabolic and functional impairment are those corresponding to the frontal and prefrontal areas while sparing the parietal and most of the temporal areas—are consistent with FTLD.

Furthermore, it is initially striking that the patients lack classical cardiovascular risk factors that would justify the remarkable WMH, predominantly frontal, found in both cases. Alternatively, these findings may indeed be related to amyloid accumulation causing myelin dysfunction, as suggested in previous studies [37,38]. In some studies with anatomopathological confirmation, it has been found that mixed cerebral pathology, of vascular and Alzheimer’s type, has a prevalence of between 20% and 38%. It is known that cerebrovascular disease is caused by CV risk factors and amyloid status [39]. Vascular brain injury and white matter hyperintensities can often be strategically placed to disrupt frontal–subcortical circuits and cause alterations in executive function. Some of these frontal–subcortical networks are important in behavior modulation. The dorsolateral prefrontal circuit regulates central executive control, which includes both anticipation and the planning process. Disturbance along this pathway can result in impairments in cognitive testing, including deficient attention as well as changes in dysexecutive function. Two other frontal–subcortical circuits that may have been affected in our both cases involved behavior: The anterior cingulate circuit mediates motivation, and the orbitofrontal circuit participates in mediating inhibition [39].

Concerning the cerebral areas most commonly affected in PSEN2-AD, no changes were found in comparison with sporadic AD, with the parietal and temporal lobes being the main targets of neurodegeneration and thus hypoperfusion and hypometabolism. However, inconsistent frontal lobe involvement has been noted in some cases [8]. For instance, in 2009, the case of a 62-year-old male carrier of a substitutional mutation in PSEN2 also made its clinical debut resembling FTLD [40]. Thus, the two new cases we present are, as far as we know, further evidence of an exceptional clinical picture of *PSEN2*. Alternatively, this clinical presentation may be far more common but systematically underdiagnosed, as PSEN2 mutations are not routinely evaluated in cases of apparent FTD. We have also tried to analyze two of the most remarkable features of our cases (the clinical FTD phenotype or the occurrence of WMH), taking into account the information collected in the Alzforum platform. So far, there are 90 mutations described. The FTD phenotype has been described in relation to mutations other than those presented in our patients, all of them with clinical onset in young patients (between 31 and 62 years of age), with both behavioral variants (Mut. T388M, exon 12) and primary language impairment (Mut. Y231, exon 8) being more precisely characterized in another case as non-fluent primary progressive aphasia (Mut. H169N, exon 7).

Regarding the presence of frontal-predominant WMH in neuroimaging in patients with other *PSEN2* mutations, similar findings have been described in a minority of cases, especially in mutations in exons 4 and 5, with reports in some of them also of increased accumulation of amyloid deposits, especially in frontal regions.

Finally, with respect to biomarkers of neurodegeneration, our results showed that both patients showed elevated p-tau and t-tau levels as well as higher concentrations of Aβ1–42, with positive results for the ratio and β-amyloid ratio (1–42/1–40) [20]. Importantly, these results demonstrate that not only is β-amyloid is dysregulated but also tau protein and p-tau.

## 4. Conclusions

We herein provide two new well-studied patients with well-founded diagnoses and, with these mutations as the only relevant genetic findings, aimed to increase the available information about them, their pathogenicity, and the potential role they play in the clinical picture of the patients.

The rare heterozygous variants in PSEN2 identified in this work, namely c.772G>A (p.Ala258Thr) and c.1073-2_1073-1del, may be genetic alterations that predispose to or cause dementia, with this clinical pattern mimicking FTD (behavioral–language) and showing congruent neuroimaging of frontal impairment, though more cases are needed to confirm this hypothesis. In the same way, we hypothesize the possibility that certain mutations in PSEN2 may directly or indirectly cause the WMH observed in these patients. This may arguably be related to amyloid accumulation causing myelin dysfunction.

Considering the possible association of PSEN2 and a clinical picture suggesting FTD, it may be reasonable to recommend the inclusion of the analysis of this gene when requesting a genetic panel in cases of suspected FTD. This approach may be especially valid in familial cases with no known FTD mutations in the first analysis.

The use of massive sequencing by broad gene panels in the study of dementia allows us to make unsuspected diagnoses, although the interpretation of variants of uncertain significance requires careful management in which the comprehensive clinical and family study is relevant for weighing the pathogenicity of such variants and for offering family genetic counseling.

### Limitations and Strengths of the Study

This study has the limitations of an observational, descriptive, single-center study as well as the biases of a systematic review (for example, the possible variability in the collection of certain data extracted from documents produced by more than one professional). The solid clinical characterization and the complete diagnostic study of the patients presented are its most important strengths.

## Figures and Tables

**Figure 1 biomedicines-12-01881-f001:**
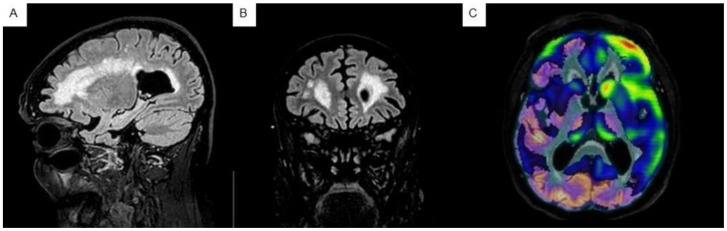
(**A**,**B**) Cerebral MRI of case A, with axial and coronal T2-FLAIR sequence displayed. (**C**) PET-FDG showing a marked frontal hypometabolism in all the slice planes used in case A. Both studies were performed in 2016.

**Figure 2 biomedicines-12-01881-f002:**
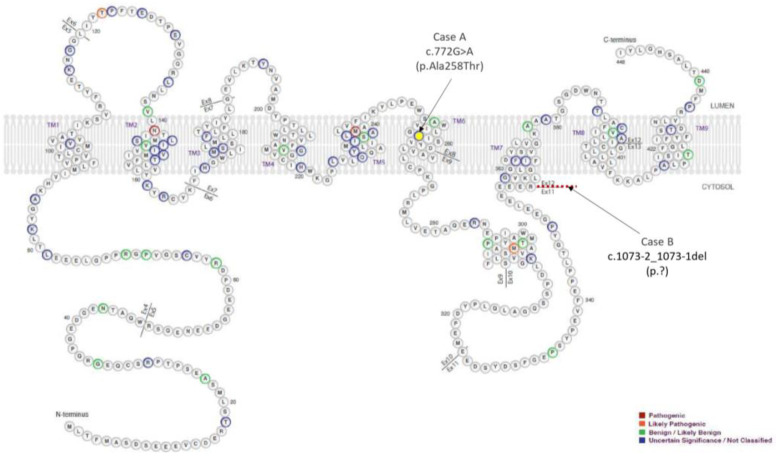
Schematic representation of full-length amino acid sequence of presenilin-2 retrieved and modified from Alzforum web (https://www.alzforum.org/, Version 3.2—2023). The colors of the variants represent the likelihood of pathogenicity. Mutations are named according to amino acid position in isoform 1 (Uniprot: P49810), which has 448 amino acids. The variants reported in the present study are indicated by arrows.

**Figure 3 biomedicines-12-01881-f003:**
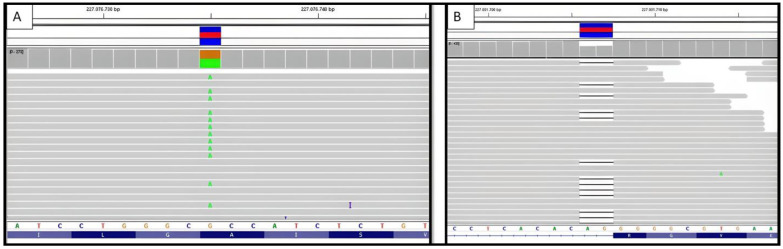
(**A**,**B**) Visualization of identified variants using the IGV. Case A: PSEN2 (NM_000447.3):C.772G>A (p.Ala258Thr). Case B: NM_000447.3:c.1073-2_1073-1del.

**Figure 4 biomedicines-12-01881-f004:**
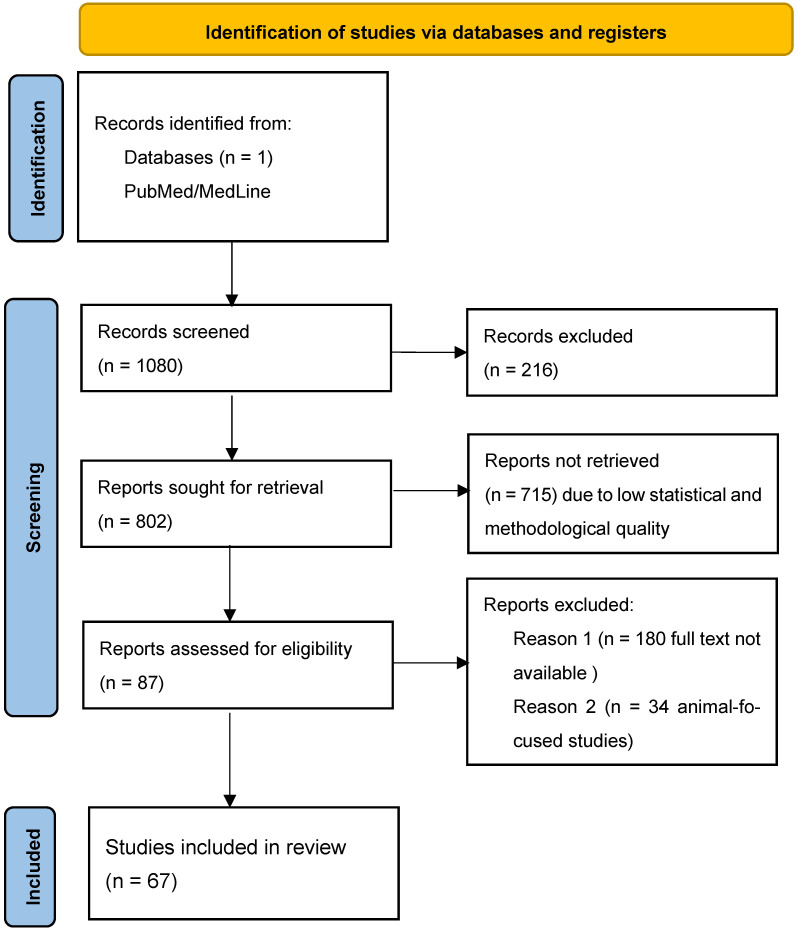
Flow chart of the literature review.

**Table 1 biomedicines-12-01881-t001:** Results of neuropsychological and functional tests performed on both patients, with a longitudinal perspective, during follow-up and evolution.

	Case B			Case A		
CDR_	26 April 2017	4 December 2018	4 June 2020	18 June 2014	22 June 2016	22 June 2018
CDR_MEMORY	0.5	2	3	1	NV *	NV
CDR_ORIENTATION	1	1	3	1	2	3
CDR_REASONING AND PROBLEM SOLVING	0	2	3	1	NV	3
CDR_OUT-OF-HOME ACTIVITIES	0.5	2	3	1	3	3
CDR_DOMESTIC ACTIVITIES AND HOBBIES	0	2	3	1	2	3
CDR_SELF-CARE	0	2	3	1	2	3
CDR_TOTAL	0.5	2	3	1	2	3
MMSE	21	8	0	8	NV	NV
NPI_TOTAL	25	34	46	12	16	25
NPI_DELUSIONS (frequencyXgravity)	×1	0	0	0	0	0
NPI_HALLUCINATIONS (frequencyXgravity)	0	0	0	0	0	0
NPI_AGITATION (frequencyXgravity)	0	0	4 × 2	0	0	1 × 1
NPI_DEPRESSION-DYSPHORIA (frequencyXgravity)	4 × 2	4 × 1	0	4 × 1	4 × 2	4 × 2
NPI_ANXIETY (frequencyXgravity)	4 × 2	4 × 1	4 × 2	4 × 1	0	1 × 1
NPI_EUPHORIA (frequencyXgravity)	0	4 × 2	4 × 2	0	0	4 × 3
NPI_APATHY (frequencyXgravity)	4 × 2	4 × 2	4 × 2	0	4 × 2	1 × 1
NPI_DISINHIBITION (frequencyXgravity)	0	4 × 2	4 × 2	2 × 2	0	1 × 1
NPI_IRRITABILITY (frequencyXgravity)	0	2 × 2	3 × 2	2 × 2 (hygiene)	0	1 × 1
NPI_ABERRANT MOTOR BEHAVIOR (frequencyXgravity)	0	0	0	0	0	0
NPI_CAREGIVER STRESS	6	8	10	3	3	3
APETITE	OK	HYPERPHAGIA	HYPERPHAGIA	HYPERPHAGIA	HYPERPHAGIA	OK
SLEEP	OK	OK with medication	OK with medication	OK	Fragmentated	Fragmentated
BARTHEL	100	90	40	100	50	20
AIVD	8	2	2	3	0	0
MIS	2/8	0/8	0	2/8	NV	NV
SEMANTIC FLUENCY	--	2	0	2	0	0
PHONETIC FLUENCY	--	0	0	0	0	0
VISOCONSTRUCTIVE PX (pentagons)	0 (JUST ONE)	0	0			
ORIENTATION	5/10	1/10	0/10	2/10	NV	NV
Boston Aphasia Severity Scale	4/5	3/5	1/5	2/5	1/5	0/5

* NV, not valuable due to language impairment. In the vertical plane, the change of the background color is intended to visually separate the two cases. In the horizonal (in the “NPI_ TOTAL” row) it has been placed to imply that subsequent sections are part of the NPI scale.

**Table 2 biomedicines-12-01881-t002:** Analysis of plasma metabolites in both cases, with the experimental cut-off points proposed by https://www.medrxiv.org/content/10.1101/2023.04.20.23288852v1 (accessed on 9 September 2023).

Cut-Off Points	None Proposed	None Proposed	<0.083	>2 pg/mL
ID	Aβ1–42	Aβ1–40	β-amyloid ratio (1–42/1–40)	p-tau
Case A	18.31	318.71	0.057	2.39
Case B	2.96	265.42	0.011	2.91

**Table 3 biomedicines-12-01881-t003:** Analysis of CSF metabolites in case A, with the medically accepted cut-off points.

Cut-Off Points	<638 pg/mL	None Proposed	<0.069	>56.5 pg/mL	>404 pg/mL
ID	Aβ1–42	Aβ1–40	β-amyloid ratio (1–42/1–40)	p-tau	t-tau
Case A	988	10,693	0.092	122.5	

## Data Availability

The original contributions presented in the study are included in the article, further inquiries can be directed to the corresponding authors.

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
