# Peer review of "PSEN2 Mutations May Mimic Frontotemporal Dementia: Two New Case Reports and a Review"

_biomedicines, 2024, doi:10.3390/biomedicines12081881_

Round 1

Reviewer 1 Report

Comments and Suggestions for Authors

Here, these authors present two new cases with PSEN2 mutations which despite the presence of Alzheimer's disease (AD) fluid biomarkers, manifested clinically as fronto-temporal dementia (FTD). The two women were aged 69 and 62 years at the clinical onset presenting with abnormal behaviour and language dysfunction which progressed to dementia over three years. MRI’s showed frontal atrophy and confluent white matter small vessel disease while FDG PET showed greater frontal than temporo-parietal hypometabolism.  NG genetic screening of the PSEN2 gene found an Ala238Thr mutation in one case and a deletion in the other. A systematic review of the PSEN2 mutations was performed which revealed other cases where mutations had led to an FTD phenotype. It is concluded that when screening FTD cases for genetic mutations  it may be reasonable to include PSEN2 in panels.

The case reports are novel and the case for PSEN2 mutations presenting as FTD despite having fluid biomarkers of AD present is well made. A difficulty is that, in the absence of post-mortem confirmation, the presence of FTD pathology cannot be excluded although a genetic screen for causative genes was negative. Additionally, it has not been proven that the Ala238Thr PSEN2 mutation is pathogenetic. Having said that, these case reports are of interest.

Comments on the Quality of English Language

The authors should avoid using words like 'notorious' and 'remarkably' as these are subjective and unnecessary.  

Author Response

Here, these authors present two new cases with PSEN2 mutations which despite the presence of Alzheimer's disease (AD) fluid biomarkers, manifested clinically as fronto-temporal dementia (FTD). The two women were aged 69 and 62 years at the clinical onset presenting with abnormal behaviour and language dysfunction which progressed to dementia over three years. MRI’s showed frontal atrophy and confluent white matter small vessel disease while FDG PET showed greater frontal than temporo-parietal hypometabolism.  NG genetic screening of the PSEN2 gene found an Ala238Thr mutation in one case and a deletion in the other. A systematic review of the PSEN2 mutations was performed which revealed other cases where mutations had led to an FTD phenotype. It is concluded that when screening FTD cases for genetic mutations  it may be reasonable to include PSEN2 in panels.

The case reports are novel and the case for PSEN2 mutations presenting as FTD despite having fluid biomarkers of AD present is well made. A difficulty is that, in the absence of post-mortem confirmation, the presence of FTD pathology cannot be excluded although a genetic screen for causative genes was negative. Additionally, it has not been proven that the Ala238Thr PSEN2 mutation is pathogenetic. Having said that, these case reports are of interest.

Response:

We thank you for your words and for your opinions and suggestions. 

Certainly, no post-mortem studies were performed on the deceased patient, so the biomarkers in CSF and the negative genetics in the extensive genetic panel for FTD is what we are relying on.

Indeed, we are talking about variants already described in the Alzforum platform. Many of the mutations listed here lack clinical information or biological effect. Also, in many cases the pathogenicity has not been demonstrated conclusively, being sometimes a probability based on consistency with the clinical phenotype, comparison with other pathogenic mutations in this gene and other factors. In none of the mutations presented in our patients has there been a definitive consensus about their pathogenicity.

In case A, for example, the CADD tool, which integrates diverse information in silico, predicted a damaging effect, with a PHRED-scaled CADD score of 21.2, slightly above 20, a commonly used threshold for assessing deleteriousness. In case B, the mutation had a CADD-PHRED score of 23.9, suggesting it is in the top 1 percent of deleterious variants, although in the only existing report it was described in patients with variable phenotypes.

Our intention is to provide two new well-studied patients with well-founded diagnoses and with these mutations as the only relevant genetic findings, trying to increase the available information about them, their pathogenicity and the potential role they play in the clinical picture of the patients.

We have removed and changed terms like "notorious" and "remarkably".

Thanks once more.

Reviewer 2 Report

Comments and Suggestions for Authors

This is a clinical presentation and literature review of two subjects who presented with early onset dementia, were felt to have some form of frontal lobar dementia, were extensively evaluated and found to have specific mutations (one with deletions) in the presenilin-2 (PSEN2) gene. Autosomal dominant causes of Alzheimer's disease (AD) have nearly complete penetrance (the authors state 95%), are extremely rare (account for ~1% of AD cases) and involve the PSEN2 gene least often (compared to mutations in APP or PSEN1 gene). The authors provide extensive clinical histories, imaging, biochemical, neuropsychological testing (over time) and gene sequencing data on their two subjects. The first subject (patient A) died, and unfortunately no postmortem brain examination is reported. The guardian of the second patient (B) did not give permission for lumbar puncture to obtain CSF for biochemical analyses, so the reader does not have data concerning beta amyloid and phospho-tau levels in CSF. 

I found the Discussion to be a bit too long, but otherwise it was comprehensive. The authors use the abbreviation "CAA" (presumably for cerebral amyloid angiopathy), but unless I missed it, do not define "CAA" for the reader.

This is a meaningful case report which serves as an excellent teaching example for the clinical characteristics of a particularly rare form of autosomal AD. Clinical presentation and histories are now of little specific diagnostic use, and unfortunately, one is now obliged to carry out extensive biochemical, imaging and molecular genetic analyses to render this type of devastating diagnosis.

Comments on the Quality of English Language

English language is overall OK. There are several misspellings and subject-verb discrepancies, but these are easily corrected during final editing.

Author Response

This is a clinical presentation and literature review of two subjects who presented with early onset dementia, were felt to have some form of frontal lobar dementia, were extensively evaluated and found to have specific mutations (one with deletions) in the presenilin-2 (PSEN2) gene. Autosomal dominant causes of Alzheimer's disease (AD) have nearly complete penetrance (the authors state 95%), are extremely rare (account for ~1% of AD cases) and involve the PSEN2 gene least often (compared to mutations in APP or PSEN1 gene). The authors provide extensive clinical histories, imaging, biochemical, neuropsychological testing (over time) and gene sequencing data on their two subjects. The first subject (patient A) died, and unfortunately no postmortem brain examination is reported. The guardian of the second patient (B) did not give permission for lumbar puncture to obtain CSF for biochemical analyses, so the reader does not have data concerning beta amyloid and phospho-tau levels in CSF. 

I found the Discussion to be a bit too long, but otherwise it was comprehensive. The authors use the abbreviation "CAA" (presumably for cerebral amyloid angiopathy), but unless I missed it, do not define "CAA" for the reader.

This is a meaningful case report which serves as an excellent teaching example for the clinical characteristics of a particularly rare form of autosomal AD. Clinical presentation and histories are now of little specific diagnostic use, and unfortunately, one is now obliged to carry out extensive biochemical, imaging and molecular genetic analyses to render this type of devastating diagnosis.

Reponse:

Thank you very much for your words, comments and suggestions. We are very grateful that you find this work illustrative and interesting.

Indeed, the acronym CAA had been used without specifying what the initials stand for, so it has been explained.

Reviewer 3 Report

Comments and Suggestions for Authors

Dear Editor-in-Chief

Thank you for inviting me to review the manuscript entitled "PSEN2 mutations may mimic frontotemporal dementia: two new case reports and a review". In the present study, the authors provided clinical information regarding two PSEN2-AD cases with unusual clinical and neuroimaging findings. They interestingly discussed how PSEN2 mutations could be associated with FTD. Although this study addressed an interesting concept, some issues should be considered:

1. In the abstract, “Women aged 69 and 62 years at clinical onset, marked by prominent behavioral and language dysfunction, progressing to severe dementia within three years. Neuroimaging, laboratory study, genetic testing.” sounds confusing. Furthermore, I do not understand “Neuroimaging, laboratory study, genetic testing” here.

2. The role of PSEN2 mutations in FTD and its association with AD is not clearly mentioned in the abstract.

3. The paragraphs of the introduction are too high. I recommend making it more concise and reducing the number of paragraphs.

4. The problem of comment three could be observed in the discussion and conclusion sections. You ought to increase the integrity of the text and not jump rapidly from concept to concept.

5. Check the whole manuscript as some punctuation errors are present.

Comments on the Quality of English Language

Check the whole manuscript as some punctuation errors are present.

Author Response

Thank you for inviting me to review the manuscript entitled "PSEN2 mutations may mimic frontotemporal dementia: two new case reports and a review". In the present study, the authors provided clinical information regarding two PSEN2-AD cases with unusual clinical and neuroimaging findings. They interestingly discussed how PSEN2 mutations could be associated with FTD. Although this study addressed an interesting concept, some issues should be considered:

1. In the abstract, “Women aged 69 and 62 years at clinical onset, marked by prominent behavioral and language dysfunction, progressing to severe dementia within three years. Neuroimaging, laboratory study, genetic testing.” sounds confusing. Furthermore, I do not understand “Neuroimaging, laboratory study, genetic testing” here.

2. The role of PSEN2 mutations in FTD and its association with AD is not clearly mentioned in the abstract.

3. The paragraphs of the introduction are too high. I recommend making it more concise and reducing the number of paragraphs.

4. The problem of comment three could be observed in the discussion and conclusion sections. You ought to increase the integrity of the text and not jump rapidly from concept to concept.

5. Check the whole manuscript as some punctuation errors are present.

Response:

We are very grateful for your comments, impressions and suggestions. 

We have carried out a complete analysis of the manuscript following your notes, both in the formal and content aspects. We have restructured some paragraphs in different sections in order to achieve greater coherence and message consistency. In addition, some items were removed from the Introduction section in order to make it more concise and visually pleasing.

Our main intention is to provide two new well-studied patients with well-founded diagnoses and with these mutations as the only relevant genetic findings. This is why we wanted to characterize in such detail, both in the patient descriptions and in the results. We found several specific topics of special interest, such as the role of CAA, seizure presence or microvascular disease in these PSEN2-AD patients, believing that our meticulous work can make a contribution in this regard.

We have added some changes in the abstract, making it clearer (eliminating the phrase "Neuroimaging, laboratory study, genetic testing") and more reliable.

Thank you once more

At your disposal.